# A Two-Stage Low-Altitude Remote Sensing Papaver Somniferum Image Detection System Based on YOLOv5s+DenseNet121

Qian Wang [1,2,3], Chunshan Wang [1,2,3,*], Huarui Wu [2], Chunjiang Zhao [2], Guifa Teng [1,3], Yajie Yu [4] and Huaji Zhu [2]

1   School of Information Science and Technology, Hebei Agricultural University, Baoding 071001, China; hebau_wangqian@163.com (Q.W.); tguifa@hebau.edu.cn (G.T.)
2   National Engineering Research Center for Information Technology in Agriculture, Beijing 100097, China; wuhr@nercita.org.cn (H.W.); zhaocj@nercita.org.cn (C.Z.); zhuhj@nercita.org.cn (H.Z.)
3   Hebei Key Laboratory of Agricultural Big Data, Baoding 071001, China
4   School of Software, Tsinghua University, Beijing 100084, China; yuyajie130@mail.tsinghua.edu.cn
*   Correspondence: wangcs007@nercita.org.cn

**Abstract:** *Papaver somniferum* (opium poppy) is not only a source of raw material for the production of medical narcotic analgesics but also the major raw material for certain psychotropic drugs. Therefore, it is stipulated by law that the cultivation of *Papaver somniferum* must be authorized by the government under stringent supervision. In certain areas, unauthorized and illicit *Papaver somniferum* cultivation on private-owned lands occurs from time to time. These illegal *Papaver somniferum* cultivation sites are dispersedly-distributed and highly-concealed, therefore becoming a tough problem for government supervision. The low-altitude inspection of *Papaver somniferum* cultivation by unmanned aerial vehicles has the advantages of high efficiency and time saving, but the large amount of image data collected needs to be manually screened, which not only consumes a lot of manpower and material resources but also easily causes omissions. In response to the above problems, this paper proposed a two-stage (target detection and image classification) method for the detection of *Papaver somniferum* cultivation sites. In the first stage, the YOLOv5s algorithm was used to detect *Papaver somniferum* images for the purpose of identifying all the suspicious *Papaver somniferum* images from the original data. In the second stage, the DenseNet121 network was used to classify the detection results from the first stage, so as to exclude the targets other than *Papaver somniferum* and retain the images containing *Papaver somniferum* only. For the first stage, YOLOv5s achieved the best overall performance among mainstream target detection models, with a Precision of 97.7%, Recall of 94.9%, and mAP of 97.4%. For the second stage, DenseNet121 with pre-training achieved the best overall performance, with a classification accuracy of 97.33% and a Precision of 95.81%. The experimental comparison results between the one-stage method and the two-stage method suggest that the Recall of the two methods remained the same, but the two-stage method reduced the number of falsely detected images by 73.88%, which greatly reduces the workload for subsequent manual screening of remote sensing *Papaver somniferum* images. The achievement of this paper provides an effective technical means to solve the problem in the supervision of illicit *Papaver somniferum* cultivation.

**Keywords:** *Papaver somniferum* inspection; unmanned aerial vehicle (UAV); small target detection; YOLOv5s; two-stage detection and classification

## 1. Introduction

*Papaver somniferum* (opium poppy) is an annual herb cultivated in many parts of Asia. On the one hand, the *Papaver somniferum* seed is an important food product, which contains amino acids such as glycine, alanine, tyrosine, phenylalanine, and aspartic acid, which are beneficial to health. It is widely used in making all kinds of baked foods and salads. On

the other hand, *Papaver somniferum* fruit extracts are the source of a variety of sedatives, such as morphine, thebaine, codeine, papaverine, and noscapine [1]. In particular, *Papaver somniferum* fruit is the major raw material for the production of opium. Thus, it is stipulated by law in many countries and regions that the cultivation and sales of *Papaver somniferum* must be authorized by the government and are subject to official supervision throughout the entire industry chain.

Despite stringent legal constraints, there are still people who cultivate *Papaver somniferum* in private fields or yards without authorization, and then sell and eat *Papaver somniferum* illegally in certain countries and regions. Illicit *Papaver somniferum* cultivation causes potential harm to the social order. Once *Papaver somniferum* related products are out of regulation and flow into the market, they pose a serious threat to people's physical and mental health. Hence, it has become an urgent issue for local regulatory authorities to investigate illicit *Papaver somniferum* cultivation. In view of this, the rapid and accurate detection of illicit *Papaver somniferum* cultivation sites is an important means of combating such crimes. Nakazawa et al. [2] identified the differences in spectral characteristics between opium poppy and other crops by utilizing hyperspectral imaging and developed a method to detect illicit opium poppy fields based on hyperspectral data. However, hyperspectral instruments are generally expensive, so the rapid detection of opium poppy from remote-sensing images is not the first choice for researchers. At present, the use of unmanned aerial vehicle (UAV) equipped with high-definition cameras to inspect illicit *Papaver somniferum* cultivation is deemed an accurate and effective approach. Zhou et al. [3] and Wang et al. [4] realized the detection of low-altitude remote sensing opium poppy images by applying a target detection algorithm. Nevertheless, in large-scale high-resolution remote sensing images, the accurate detection of opium poppy is actually a very challenging task due to a variety of problems (e.g., blurred images due to natural shooting conditions; existence of easily-confused objects such as green onion, white flowers, wine bottle, plastic toy, green plastic film, etc.; complex backgrounds; etc.) Currently, the complete process of opium poppy screening and elimination is as follows: (1) after the UAV captures the images, anti-drug experts will conduct a naked eye search on a large number of collected images to label and record the suspicious targets in the images; (2) the relevant government departments will collect coordinate points for the suspicious targets and then visit the sites to uproot the *Papaver somniferum* plants. Although this method can accurately locate opium poppy cultivation sites, it has many obvious shortcomings. For example, the number of images collected by UAV can be incredibly huge, which is often counted on the order of terabytes. Screening of these images relying on the naked eye is extremely laborious and time-consuming, and the accuracy of the results is highly dependent on the proficiency and skills of the inspectors.

With the rapid development of artificial intelligence, target detection algorithms have been successfully applied to various fields such as medicine [5], agriculture [6], forestry [7], physics [8], and remote sensing communities [9].The target detection algorithm can be traced back to the cascade classifier framework proposed by in 2001 [10], which brought target detection into reality for the first time. In 2005, Dalal N. and Triggs B [11] achieved human detection using Histogram of Gradients (HOG) and Support Vector Machines (SVM). Early scholars mainly relied on expert experience to describe and extract the shape, color, texture, and other features of objects and then to implement target detection using artificial feature extraction algorithms. Until 2012, Krizhevsky [12] innovatively applied AlexNet to the field of image classification [13]. Since then, researchers have turned their attention from artificial feature extraction methods to deep learning methods. The core of deep learning lies in the fact that the feature extraction network can realize automatic target feature extraction, which greatly reduces the difficulty and improves the accuracy and efficiency of target detection.

Target detection methods based on deep learning can be generally divided into one-stage methods and two-stage methods. The two-stage method is implemented in the following steps: (1) select region proposals from the input images that may contain the

target object; (2) classify and perform location regression on the region proposals to obtain the detection results. The one-stage method omits the step of generating region proposals and directly computes anchor boxes of different sizes and aspect ratios in the input images. The commonly used two-stage detectors include R-CNN [14], Fast R-CNN [15], Faster R-CNN [16], Mask R-CNN [17], R-FCN [18], and Cascade R-CNN [19]. The one-stage target detection algorithms mainly include the SSD [20–24] series and the YOLO series [25–28]. Compared with two-stage detectors, the inference speed of one-stage detectors is generally faster.

The target detection technology based on deep learning has increasingly wide applications in the field of remote sensing. Xu et al. [29] proposed an improved YOLOv3 algorithm by using DenseNet (Densely Connected Network) to enhance the network performance. On the basis of the original YOLOv3 network, the improved method increased the detection scale to 4 and thereby improved the mAP from 77.10% to 88.73% on the RSOD remote sensing dataset. Especially, for the detection of small targets such as aircrafts, the mAP was increased by 12.12%, suggesting that large-scale detection mechanism can help improve the accuracy of small target detection to a certain extent. By comparing the detection performance between one-stage and two-stage detectors on a single UAS-based image, Ezzy et al. [30] concluded that the self-trained YOLOv3 detector could achieve accurate small target detection of rodent caves. In view of the difficulty in small target detection, researchers have made consistent efforts to improve the network structure from different perspectives so as to increase the accuracy of small target detection [31]. Miura et al. [32] utilized an improved CNN network and the post-earthquake aerial images in Japan to detect collapsed buildings, non-collapsed buildings, and buildings covered with blue tarps. Shi et al. [33] proposed to use the improved classic YOLOv4 model to quickly detect collapsed buildings from aerial images after the Wenchuan and Beichuan earthquakes in China and used the K-means algorithm to cluster the optimal anchor boxes from the images. By improving the CSPDarkNet53 module and the loss function in the original network, the proposed method effectively increased the average precision of collapsed building detection. Zhu et al. [34] improved the YOLOv5 algorithm based on the attention mechanism and designed a pyramid-based method for boulder detection in planetary images. The proposed network was equipped with the Convolutional Block Attention Module (CBAM) and the Efficient Channel Attention Network (ECA-Net) to improve the overall detection performance. Yan et al. [35] proposed to use the improved YOLOv5s model to distinguish apples that can be grasped from those that cannot be grasped. Specifically, the authors upgraded the CSP module to CSP-2 module and inserted the SE module in the attention mechanism network into the modified backbone network. The improved model was more accurate, faster, and lighter than the original YOLOv5s and other detection models. Mehdi et al. [36] improved yolov3 to detect of hazardous and noxious substances of vital transportation by sea. It can be seen from the research status described above that, in the field of remote sensing image detection, the one-stage YOLO series is more in line with the real-time and accurate detection requirements at the current stage.

YOLOv5 is the latest version of the YOLO series of networks and is also the most advanced network of the one-stage method. YOLOv5 treats target detection as a regression problem so as to perform localization and classification concurrently to achieve fast and accurate detection. Wang et al. [4] proposed an improved YOLOv3 algorithm, which realized fast and accurate low-altitude remote sensing opium poppy image detection. Their algorithm effectively improved the accuracy of opium poppy detection and reduced the parameter size of the model, but they did not provide a good solution for distinguishing easily confused plants and objects and for the accurate detection of opium poppy cultivation. Hu et al. [37] relied on the two-stage idea to realize the identification of individual cows. First, the YOLO algorithm was used to locate the parts of each cow of interest, and then the segmented parts were sent to three convolutional neural networks for feature extraction. After the fusion of parts features, the SVM was used to achieve individual cow identification.

In order to address the problems of labor intensity and time-consuming of manual screening opium poppy inspection images, this paper proposes a two-stage automatic detection method of "detection + classification". YOLOv5 was chosen as the first-stage detector in our study considering its characteristics of high detection accuracy and real-time performance, for the purpose of improving the recall and detection speed. Then, the detection results from the first stage were input into the second stage network for further classification, so as to improve the overall accuracy of *Papaver somniferum* detection.

## 2. Materials and Methods

### 2.1. Data Acquisition

In mainland China, the best time to plant and harvest *Papaver somniferum* is from March to August. When an UAV is used to capture images, the flying height is usually set between 130 m and 145 m in order to avoid the interference of uncertain factors such as power lines, buildings, and birds. The dataset used in the present study is composed of data obtained in 2019, 2020, and 2021, and the UAV model is fixed-wing UAV with a wingspan of 2.4 m, which is equipped with a Sony a7RIV camera. Samples of the original images collected are shown in Figure 1, and the metadata of the images are shown in Table 1. Photoshop was used to crop the original images containing papaver somniferum. Since the areas occupied by the *Papaver somniferum* in the original images are different, in order to ensure the complete appearance of the *Papaver somniferum* plot, the width and height of the cropped image are set to 640 × 640 pixels and 800 × 800 pixels, respectively. Finally, a total of 643 images were labelled. The process of labeling data is that the professionals manually screen these images and select all the images of suspected papaver somniferum, and finally, the supervisors go to the scene according to the GPS information to verify whether the images of suspected opium poppy are real papaver somniferum. If it is a real *Papaver somniferum* target, it is marked as a ground truth sample.

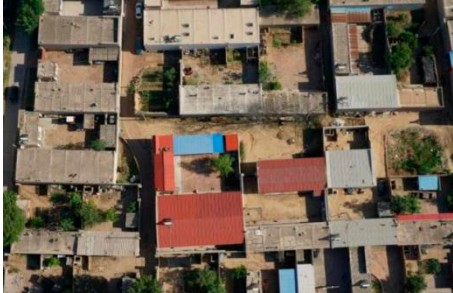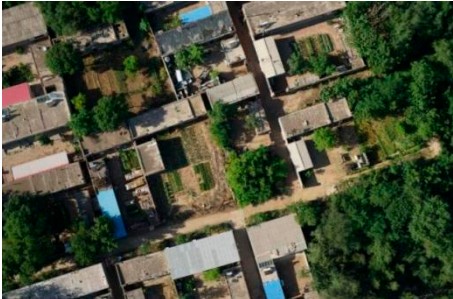

**Figure 1.** Low-altitude remote sensing images in the village area captured by UAV.

**Table 1.** The metadata of the image.

| Image File Name | Resolution | Ground Resolution/cm | File Type |
|---|---|---|---|
| DSC000001 | 7952 × 5304 | 2 | JPG |

### 2.2. Data Processing

The collection of *Papaver somniferum* images is very challenging, and the images are often affected by light, so it is difficult to accurately extract features from the images. Meanwhile, easily-confused plants, single-plants, and complex backgrounds may also negatively impact the detection precision. Therefore, appropriate image enhancement strategies need to be applied to improve the quality, color and contrast of the original images. In this paper, three enhancement methods were adopted, which are contrast enhancement, brightness enhancement, and random angle rotation with brightness enhancement. The contrast and color enhancement can simulate different weather and light conditions during image acquisition and can effectively enhance the images of opium poppy planted in the

shaded area. The random angle rotation with brightness enhancement does not change the image itself but only alters the position and direction of the image so as to increase the richness of the target. When UAV collects images, it is necessary to set a certain side overlap between routes to ensure that the image acquisition can completely cover the whole survey area. Although the overlapping part contains the ground at the same position, the ground at the same position is not exactly the same in the two images. The ground at the same position often has a certain deflection angle in the two images. Therefore, random angle rotation enhancement is taken as image data augmentation. After enhancing the contrast by 100%, enhancing the color by 100%, and enhancing the rotation angle with brightness by 200%, the total number of enhanced images is 3215. The images were labelled using the LabelImg tool, and an xml file containing the location information was generated. Table 2 presents the original and enhanced images for different planting strategies.

**Table 2.** Examples of preprocessed *Papaver somniferum* images for different planting strategies.

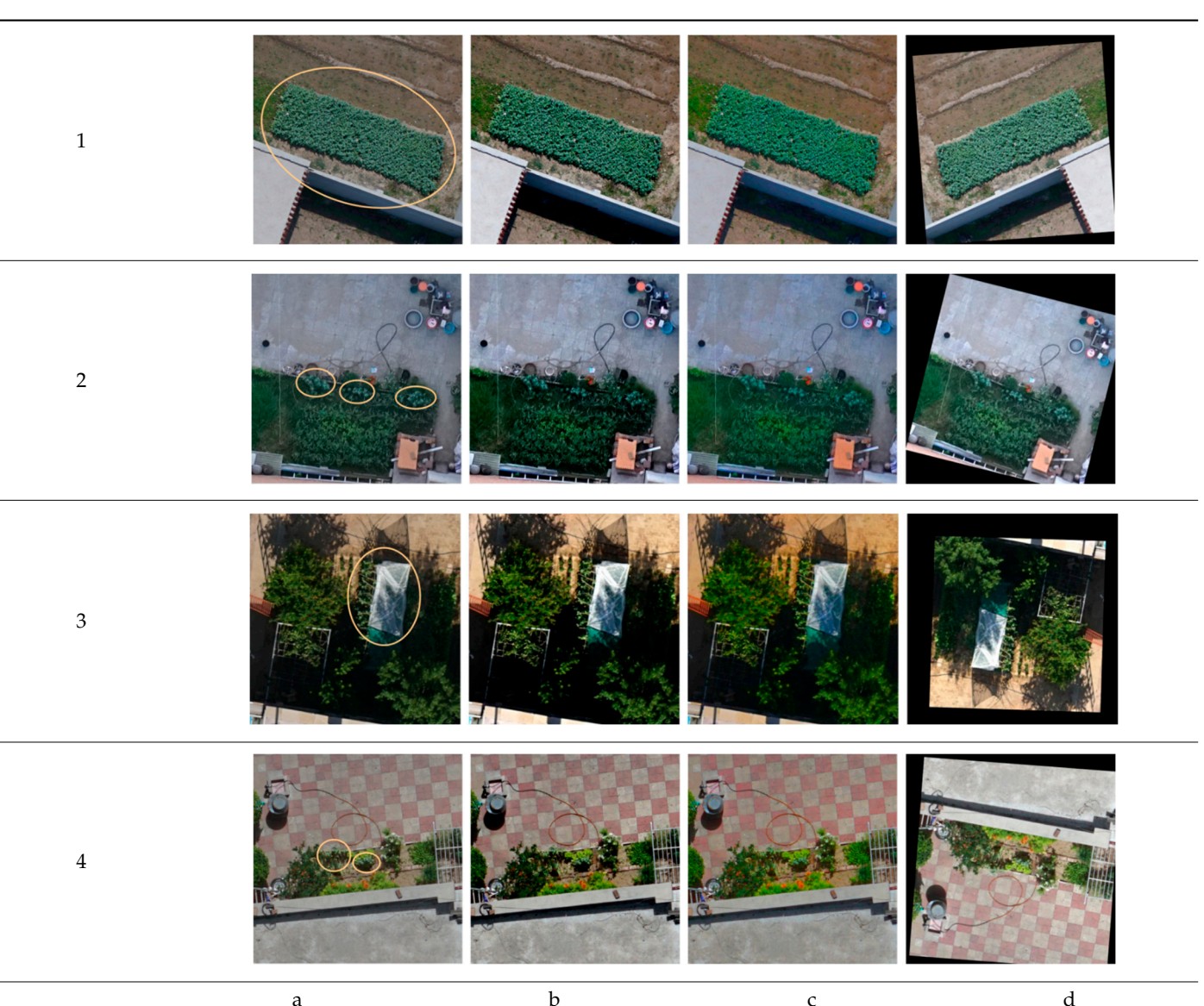

1–4 on the left represent different planting strategies (1, large-scale planting; 2, small-scale planting; 3, shaded planting; 4, mixed planting); a, b, c, d represent the original image, contrast enhancement, color enhancement, and random rotation with brightness enhancement, respectively. The yellow circle in the original image refers to the specific location of the *Papaver somniferum*.

In the classification task, the opium poppy category was from the ground truth cropping of the 643 labeled images in the original dataset. The cropped images were resized to 224 × 224 pixels. The enhancement methods included brightness enhancement, contrast enhancement, and flip enhancement. The name for the opium poppy category is "*Papaver somniferum*". The other two categories are images that are falsely detected or wrongly detected by the detector, namely, "plant" (e.g., weeds and other plants) and "ground" (e.g., roof, green field and land). Figure 2 shows the classification dataset, and Table 3 presents the specific number of data items in the classification dataset.

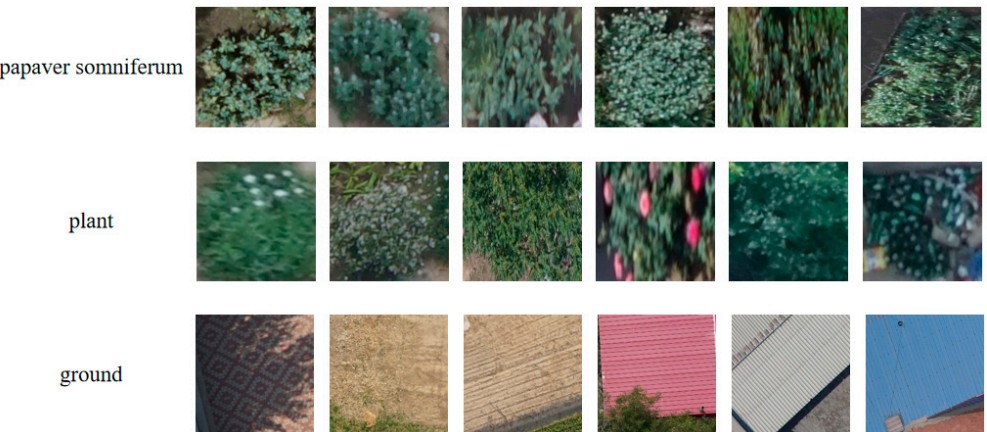

**Figure 2.** Classification dataset.

**Table 3.** The specific number of data items in the dataset.

| Category | Number of Training Images | Number of Test Images | Total |
|---|---|---|---|
| *Papaver somniferum* | 927 | 379 | 1306 |
| *plant* | 736 | 315 | 1051 |
| *ground* | 787 | 337 | 1124 |

## 3. Experiment

### 3.1. YOLOv5 Network Structure

The YOLOv5 network is mainly composed of three parts, the backbone network (Backbone), the neck network (Neck), and the output prediction part (Prediction). The Backbone functions as the main feature extraction module of the entire network, which utilizes the convolutional neural network to perform feature map convolution calculation so as to generate four feature maps of different sizes. The Backbone adopts the Focus structure. The main idea of the Focus structure is to crop the input image through the Slice operation. The Slice and Concat operations can reduce the size of the feature map by half to better extract the feature information. The Neck fuses the corresponding feature maps that are generated for the purpose of linking the context information to reduce information loss. In the fusion process, the feature pyramid structure of FPN+PAN is adopted. The FPN structure is a top-down feature map that fuses a high-level feature information with low-level features through up-sampling for prediction. It can enhance the semantic information of the pyramid. On the other hand, the PAN structure transfers the positioning feature information from the bottom to the top, so as to aggregate the parameters of different feature layers from different Backbone layers.

In the overall structure of YOLOv5, the CBL module serves as the basic module, which consists of the convolution module, the normalization module, and the Leaky relu activation function [38]. Two different Cross-Stage Partial (CSP) networks [39] are used in the Backbone and Neck network, respectively. The CSP network connects the front and rear layers of the network through cross-stage connection. After fusing shallow features with deep features, the semantic information of key areas in the image can be better transmitted,

which reduces the size of the model and improves the inference speed while ensuring the detection precision. The network structure of YOLOv5 is shown in Figure 3.

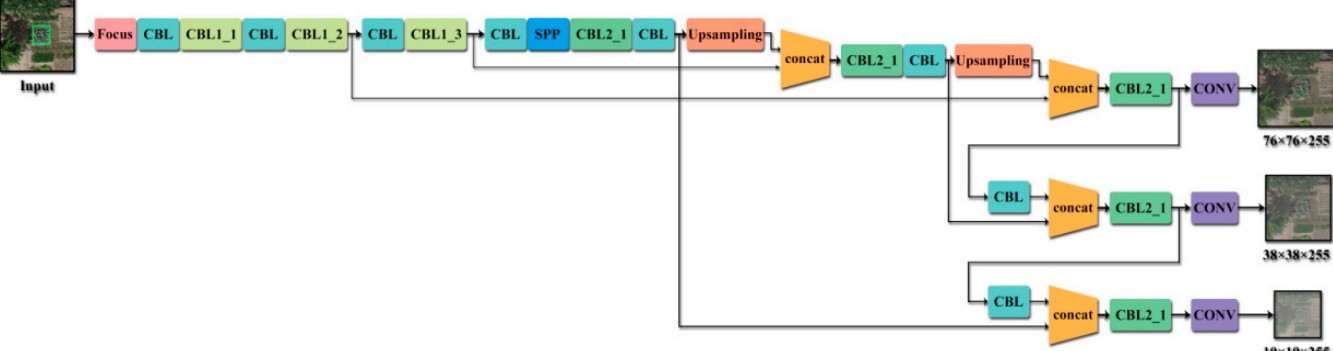

**Figure 3.** Network structure of YOLOv5.

Target detection networks with different structures have their respective advantages when dealing with different detection tasks, and most of these networks are composed of the convolutional layer, pooling layer, and fully connected layer. In our experiment, the YOLOv5 detection network delivered the best comprehensive performance and was therefore chosen for the detection task. Given the image $I$, the image label $L$ and the real coordinates $X_1$, $X_2$, $X_3$, and $X_4$ as the network input, the output of the first stage detection $[Y_1, Y_2, Y_3, Y_4, C, P]_{img}$ can be obtained through the Backbone, the Neck, and the Prediction, as shown in Equation (1).

$$[Y_1, Y_2, Y_3, Y_4, C, P]_{img} = D(I, L, X_1, X_2, X_3, X_4) \tag{1}$$

where $D(\cdot)$ is the detect ion function of YOLOv5; $C$ is the target category; $P$ is the target confidence; $Y_1$, $Y_2$, $Y_3$, and $Y_4$ represent the coordinates of the output.

*3.2. Principle of Classification Network*

The input of image classification is the original images and the image labels provided by experts. Image classification networks are mostly composed of the convolutional layer, the pooling layer, the batch normalization layer, the activation function, and the fully-connected layer, and finally, the Softmax function is used to obtain the image label probability. Different classification network structures have different feature extraction capabilities and may show different effects in different recognition tasks. In order to extract more detailed features, this study regarded the second-stage task as a three-category (i.e., "*Papaver somniferum*", "plant", and "ground") image recognition task. The images were cropped and processed into the size of 224 × 224 pixels, and the DenseNet121 classification network [40] was used to extract features from the processed images. Given image $I$ and label $L$ as the input of DenseNet121, the output $P_n$ can be obtained through feature extraction, as shown in Equation (2).

$$P_n = S(C(I), L) \tag{2}$$

where $C(\cdot)$ represents the extraction results of the feature extraction network DenseNet121; $S(\cdot)$ is the Softmax function; $P$ represents the prediction probability of different categories; $n$ represents the category.

*3.3. Two-Stage Method*

In the *Papaver somniferum* detection task, the detection results of a single detector cannot meet the actual requirements. When the model is applied in the real environment, it must have a good fit not only for the training dataset but also for the unknown dataset. For *Papaver somniferum* detection, the data collection across different times of a year is

often limited by the weather conditions and collection equipment, so that the dispersion of the data collected from different batches raises high requirements on the generalization ability of the model. In order to improve the robustness and the generalization ability of the model, this study adopted a two-stage judgment method. In the first stage, YOLOv5s was used as the detector to detect the suspicious targets, which were resized into images of $224 \times 224$ pixels. Then, a trained classification model was used to assist with the judgment to generate the final judgment results. The two-stage network structure is shown in Figure 4.

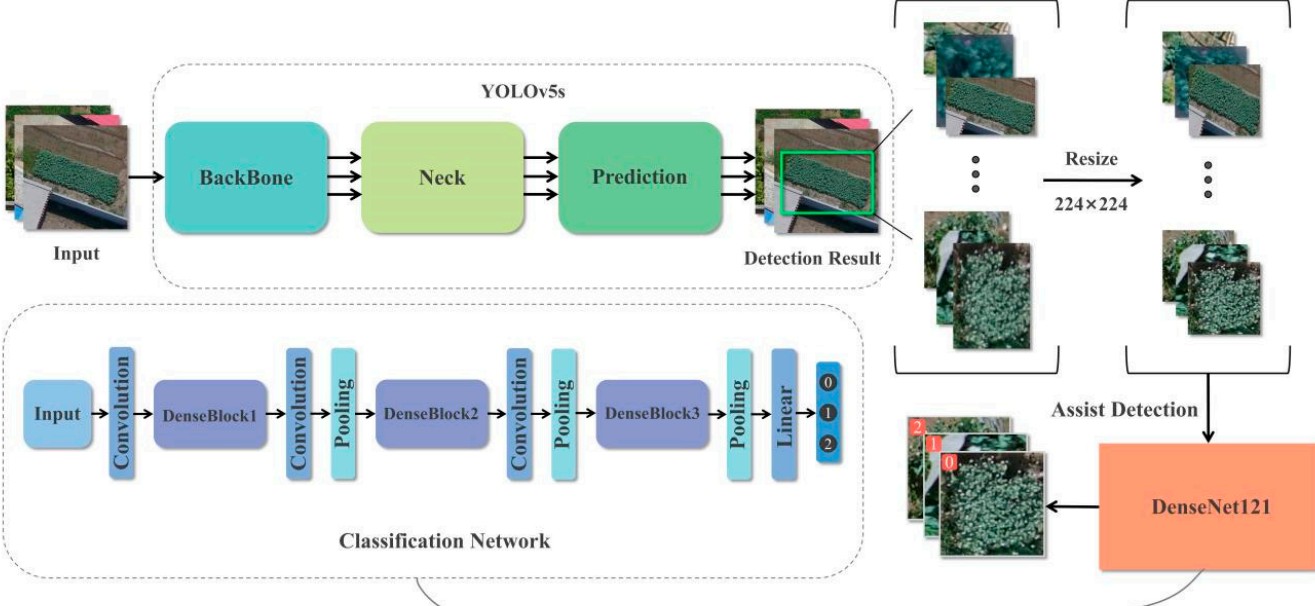

**Figure 4.** Two-stage network structure.

The input of YOLOv5s is images and the target location information; BackBone refers to the backbone network of YOLOv5s; Neck refers to the neck network of YOLOv5s; Prediction refers to the prediction stage of YOLOv5s. In the DenseNet121 classification network structure, Convolution refers to the convolutional operation; DenseBlock1, DenseBlock2 and DenseBlock3 are composed of BN, ReLu, and different numbers of $1 \times 1$ and $3 \times 3$ convolutional layers.

## 4. Experiment and Analysis

### 4.1. Experiment Dataset

The experiment in this paper consisted of two stages. In the detection task, there were a total of 643 original images, which were segmented into 3215 images after enhancement. According to the ratio of 8:2 between the training set and the test set, 2572 images were used for training, and 643 images were used for testing. The image labelling was manually processed using the LabelImg tool, and the labelling information was exported as an xml file, which contains the category information and the $x_{min}$, $y_{min}$, $x_{max}$, and $y_{max}$ information of the image. The dataset was saved in the COCO format for the training and testing purpose of the detection network model. In the classification task, the data were sorted and divided into three categories: "*Papaver somniferum*", "plant" and "ground"; the image size is $224 \times 224$ pixels.

### 4.2. Model Training

In this experiment, the hardware configuration for training and testing is as follows: Intel® Core (TM) i9-10920X CPU @ 3.00GHz, 24G memory, NVIDIA GeForce RTX 3090 graphics card, 64-bit Ubuntu 20.04.3 LTS, CUDA version 11.4, and Pytorch version 1.7.0. For the detection task, the learning rate was set to 0.01; the momentum was set to 0.937; the

weight decay was set to 0.0005; the batch_size was set to 16; and the epoch was set to 300. For the classification task, the learning rate was set to 0.0001; the optimizer adopted Adam; the batch_size was set to 16; and the epoch was set to 50.

*4.3. Evaluation Indicators*

4.3.1. Evaluation Indicators for Target Detection

The YOLOv5 model will assign a confidence threshold for each image containing the target area. In this experiment, the confidence threshold was set to 0.25. Any prediction result with the confidence threshold higher than 0.25 would be judged as a correct prediction by the model. There will be a gap between the actually labelled box and the predicted box. The smaller the gap, the more accurate the model detection results are. The Intersection over Union (IoU) is a key parameter to measure the accuracy of detection, which is expressed as Equation (3). In this experiment, IoU was set to 0.45. In Equation (3), DR stands for Detection Result, representing the prediction result; GT stands for Ground Truth, representing the actual label.

$$IoU = \frac{DR \cap GT}{DR \cup GT} \times 100\%$$ (3)

The most commonly-used indicators for target detection include Precision, Recall, F1 Harmonic Mean, Average Precision (AP), and mean Average Precision (mAP). Precision is defined as the proportion of samples that are positively labelled out of all the samples predicted to be positive. Recall is defined as the proportion of samples that are successfully predicted as positive out of all the samples that are positively labelled. Precision and Recall are calculated by Equations (4) and (5), respectively. F1, defined by Equation (6), refers to the harmonic mean of Precision and Recall. AP, defined by Equation (7), refers to the area of the closed area enclosed by the (P-R) curve. mAP, defined by Equation (8), is the main evaluation indicator of target detection, which measures the overall performance of the network. In Equation (8), $n$ represents the detection category. In this experiment, there was only one category, i.e., "*Papaver somniferum*", so $n = 1$.

$$Precision = \frac{TP}{TP + FN} \times 100\%$$ (4)

$$Recall = \frac{TP}{TP + FP} \times 100\%$$ (5)

$$F1 = \frac{2PR}{P + R}$$ (6)

$$AP = \int_0^1 P(R)dR$$ (7)

$$mAP = \frac{\sum_{i=1}^n AP_i}{n}$$ (8)

4.3.2. Evaluation Indicators for Image Classification

The performance of the second-stage classification model was evaluated by four evaluation indicators: recognition accuracy, recognition precision, model sensitivity and model specificity, which are expressed by Equations (9)–(12), respectively.

$$Accuracy = \frac{TP + TN}{TP + TN + FP + FN} \times 100\%$$ (9)

$$Precision = \frac{TP}{TP + FP} \times 100\%$$ (10)

$$Sensitivity = \frac{TP}{TP + FN} \times 100\%$$ (11)

$$Specificity = \frac{TN}{FP + TN} \times 100\% \tag{12}$$

where TP refers to the number of samples that belong to category C and are correctly classified by the classifier; FP refers to the number of samples that do not belong to category C but are misclassified by the classifier; TN refers to the number of samples that do not belong to category C and are correctly classified by the classifier; FN refers to the number of samples that belong to category C but are misclassified by the classifier.

### 4.4. Comparison and Analysis of Different Target Detection Models

The YOLOv5 algorithm consists of four different model sizes, i.e., YOLOv5s, YOLOv5m, YOLOv5l, and YOLOv5x, which are used to address different problems and different application scenarios. Table 4 presents and compares the testing results of different model structures on the same dataset, and the network structure delivering the best comprehensive performance was chosen as the final detector.

**Table 4.** Comparison between different YOLOv5 models.

| Model | Precision/% | Recall/% | F1/% | mAP/% | Total Parameter |
|-------|-------------|----------|------|-------|-----------------|
| YOLOv5s | 97.7 | 94.9 | 96.3 | 97.4 | 7,022,326 |
| YOLOv5m | 98.6 | 92.8 | 95.6 | 96.3 | 20,871,318 |
| YOLOv5l | 98.1 | 92.2 | 95.1 | 96.0 | 46,138,294 |
| YOLOv5x | 97.7 | 94.0 | 95.8 | 96.7 | 86,217,814 |

By testing the four models of YOLOv5 with different sizes and structures, it can be seen that YOLOv5m had the highest Precision, and its mAP value was 0.963. However, in the *Papaver somniferum* detection task, the higher the Recall, the better the model is. Therefore, out of the four YOLOv5 structures, YOLOv5s, which had the highest Recall, was chosen as the detector for the first stage. The mAP value of YOLOv5s was 0.974, and its overall performance was the best.

The original YOLOv5s structure was used for the training and testing of the first stage. Table 5 presents the comparison results between YOLOv5s and other existing networks in terms of the comprehensive performance of P, R, F1, mAP, and the parameter size.

**Table 5.** Comparison between different target detection algorithms.

| Model | Precision/% | Recall/% | F1/% | mAP/% | Total Parameter |
|-------|-------------|----------|------|-------|-----------------|
| Faster rcnn | 49.8 | 66.4 | 57.0 | 62.2 | 137,078,239 |
| Mask-RCNN | 70.7 | 87.7 | 78.3 | 82.6 | 63,733,406 |
| RetinaNet | 88.8 | 64.8 | 75.0 | 75.8 | 37,968,692 |
| CenterNet | 98.6 | 60.3 | 74.8 | 84.8 | 32,665,432 |
| SSD | 94.7 | 66.0 | 78.0 | 76.8 | 26,151,824 |
| YOLOv3 | 97.7 | 91.9 | 95.0 | 93.3 | 61,949,149 |
| YOLOv4 | 96.0 | 86.3 | 91.0 | 89.3 | 64,363,101 |
| YOLOv5s | 97.7 | 94.9 | 96.3 | 97.4 | 7,022,326 |

As can be seen from Table 5, YOLOv5s delivered the best results in terms of Recall, F1 score and mAP. The Precision, Recall and mAP of Faster RCNN were all at a low level. The Precision of other detection algorithms was generally higher than 88%. The Recall of RetinaNet, CenterNet, and SSD were lower than 70%, while that of YOLOv3 and YOLOv5s were higher than 90%. Based on the comparison between mainstream target detection algorithms, it was found that YOLOv5s delivered the best comprehensive performance with stronger robustness, so it is a more suitable option for solving practical problems. In order to further analyze the robustness of different models, different *Papaver somniferum* cultivation strategies were selected from the test results for comparative analysis, and the results are presented in Table 6.

**Table 6.** Detection results.

| Model | Detection Result | | | |
|---|---|---|---|---|
| | **Sample 1** | **Sample 2** | **Sample 3** | **Sample 4** |
| Faster RCNN | | | | |
| Mask-RCNN | | | | |
| RetinaNet | | | | |
| CenterNet | | | | |
| SSD | | | | |
| YOLOv3 | | | | |
| YOLOv4 | | | | |
| YOLOv5s | | | | |

From the comparison results between different detection algorithms, it can be found that YOLOv3, YOLOv4, and YOLOv5s successfully identified all the targets, while Faster

RCNN, RetinaNet, and SSD failed to detect single-plant *Papaver somniferum* in test picture 1. The main reason for the missed detection is attributed to the small size of the target, making it indistinguishable from the green field. In general, the detector had a weak learning ability for small targets in the feature extraction process. Mask R-CNN successfully detected the *Papaver somniferum* target, but there was a false detection in the test sample 2. The color of the false detection target is similar to that of a *Papaver somniferum*. The detection algorithms in the control group showed different abilities in detecting *Papaver somniferum* in different environments. Specifically, Faster RCNN and RetinaNet had relatively low confidence in the detection task, which conforms to the result of mAP as shown in Table 5.

Practice shows that the detection accuracy is positively correlated with the area of *Papaver somniferum* cultivation plot. On the existing data sets, the relationship between the detection accuracy and the area of *Papaver somniferum* cultivation plot is shown in Table 7.

**Table 7.** The detection accuracy and area of *Papaver somniferum* cultivation field.

| Type of Plot | Detection Accuracy/% | Area of Field S/m$^2$ |
|---|---|---|
| Large area plot | 98.4 | $S > 9$ |
| Medium area plot | 97.5 | $1 \leq S \leq 9$ |
| Small area plot | 80.0 | $S < 1$ |

It can be seen from Table 7 that the detection accuracy of *Papaver somniferum* plots planted in large and medium areas remains at a high level, but the detection accuracy of opium poppy plots planted in small areas is low. This is mainly because the characteristics of poppies in small plots are ambiguous, so detection is difficult. In addition, it is also related to the lower number of small area plot image samples in the dataset.

### 4.5. Comparison between Different Classification Models

In this section, VGG16, ResNet18, ResNet50, DenseNet121, DenseNet169, MobileNet, AlexNet, EfficientNet, Xception, and InceptionResNetv2 were chosen as the control group. The training accuracy curves and loss curves are shown in Figure 5a, and the comparison of test results is presented in Table 8. In order to confirm whether overfitting occurred due to too few data samples during the training process on the single-image branch, network retraining was performed for all the networks on the basis of pre-training of the ImageNet dataset, and the training accuracy curves and loss curves with the pre-training weight were obtained, as shown in Figure 5b. The comparison of test results is presented in Table 9.

As shown in Table 9, DenseNet121 derived the best results in terms of Accuracy, Precision, Sensitivity, and Specificity. It can be found that the comprehensive performance of the model using residual module and dense connection is at a high level. The characteristics of these two methods are the combination of shallow semantic features and deep semantic features in feature extraction, which enhances the feature extraction ability of the network. Both MobileNet and EfficientNet are lightweight networks, but the comprehensive performance gap is large. MobileNet network contains deep separable convolution. This convolution operation can greatly reduce the number of parameters while ensuring accuracy. Deep separable convolution is also used in Xception network structure, so its performance is better than EfficientNet. Its confusion matrix is shown in Figure 6. It can be seen that category 1 (plant) had the highest error rate. Our experiment showed that category 1 had the highest error rate in other control networks as well. From the original test set, it was found that the misclassified images were all green plants, which are very similar to the shape and color of papaver somniferum. According to Figure 5a,b, after retraining the model on the *Papaver somniferum* dataset on the basis of ImageNet pre-training, the accuracy and loss values can be largely optimized as compared to the initial state. The results in Tables 8 and 9 also show that the test accuracy had been greatly improved. On the one hand, it proves that a sufficiently large sample size allowed the classifier to fully learn the features of papaver somniferum. On the other hand, it also proves that overfitting did not occur when using the image classifier.

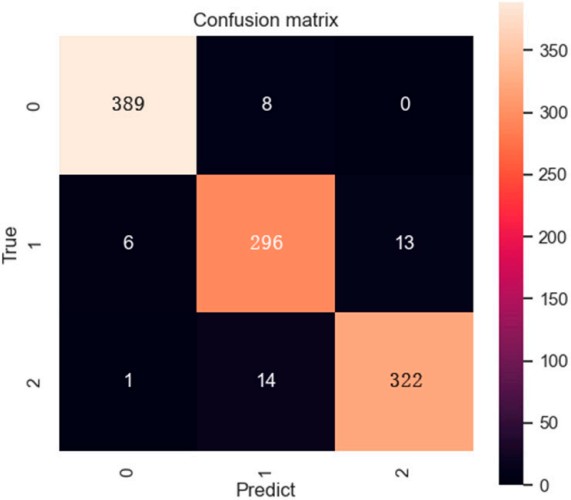

**Figure 5.** Comparison of training accuracy and loss in the classification control group.

**Figure 6.** The recognition result confusion matrix of DenseNet121.

**Table 8.** Comparison of results on image classification without pre-training in the control group.

| Models | Accuracy (%) | Precision (%) | Sensitivity (%) | Specificity (%) |
|---|---|---|---|---|
| VGG16 | 89.77 | 85.41 | 83.71 | 92.07 |
| ResNet18 | 90.59 | 85.80 | 85.62 | 92.85 |
| ResNet50 | 88.43 | 82.73 | 81.96 | 91.11 |
| DenseNet121 | 88.43 | 82.72 | 82.17 | 91.20 |
| DenseNet169 | 88.69 | 83.65 | 82.18 | 91.29 |
| MobileNet | 88.18 | 82.49 | 81.37 | 90.90 |
| AlexNet | 89.77 | 85.43 | 83.71 | 92.06 |
| EfficientNet | 89.26 | 84.07 | 83.36 | 91.76 |
| Xception | 90.59 | 86.05 | 85.39 | 92.79 |
| InceptionResNetv2 | 89.51 | 84.72 | 83.50 | 91.89 |

**Table 9.** Comparison of results on image classification with pre-training in the control group.

| Models | Accuracy (%) | Precision (%) | Sensitivity (%) | Specificity (%) |
|---|---|---|---|---|
| VGG16 | 93.52 | 90.18 | 90.18 | 95.17 |
| ResNet18 | 96.57 | 94.68 | 94.75 | 97.46 |
| ResNet50 | 96.00 | 93.89 | 93.75 | 96.99 |
| DenseNet121 | 97.33 | 95.81 | 95.83 | 98.03 |
| DenseNet169 | 97.01 | 95.45 | 95.22 | 97.75 |
| MobileNet | 96.44 | 94.46 | 94.46 | 97.36 |
| AlexNet | 92.88 | 89.29 | 89.08 | 94.60 |
| EfficientNet | 89.26 | 84.07 | 83.36 | 91.76 |
| Xception | 95.36 | 92.80 | 92.81 | 96.58 |
| InceptionResNetv2 | 97.20 | 95.59 | 95.70 | 97.96 |

In Figure 6, "0" represents "*Papaver somniferum*", "1" represents "plant", and "2" represents "ground".

### 4.6. Comparison between One-Stage Method and Two-Stage Method

This section compares the detection performance between the one-stage method and two-stage method. A total of 40 original remote sensing images were selected, consisting of 18 images containing *Papaver somniferum* and 22 images not containing papaver somniferum. The images were segmented into the size of $640 \times 640$ pixels, and 4920 images were obtained after segmentation, including 4900 images without *Papaver somniferum* and 20 images with papaver somniferum. The segmented images were input into the trained model for testing, and the prediction results were compared with the actual results. Among the 4920 images, the one-stage and two-stage methods both detected 19 images containing *Papaver somniferum* (Recall, 95%). The one-stage method had 291 false detections, and the two-stage method had 76 false detections, suggesting a reduction of false detection by 73.88%. The number of test images and the detection results are presented in Table 10. Figure 7a,b show the detection results of the one-stage method and the two-stage method, respectively. In the figure, True means correct detection, False means wrong detection.

**Table 10.** Comparison of results between one-stage and two-stage methods.

| Model | Total Number of Original Remote Sensing Images | | Total Number of Image Patches after Segmentation | | Detection Result | |
|---|---|---|---|---|---|---|
| | **40** | | **4920** | | Correct Detection | Wrong Detection |
| | Contain *Papaver somniferum* | No *Papaver somniferum* | Contain *Papaver somniferum* | No *Papaver somniferum* | | |
| YOLOv5s | 18 | 22 | 20 | 4900 | 19 | 291 |
| YOLOv5s+DenseNet121 | 18 | 22 | 20 | 4900 | 19 | 76 |

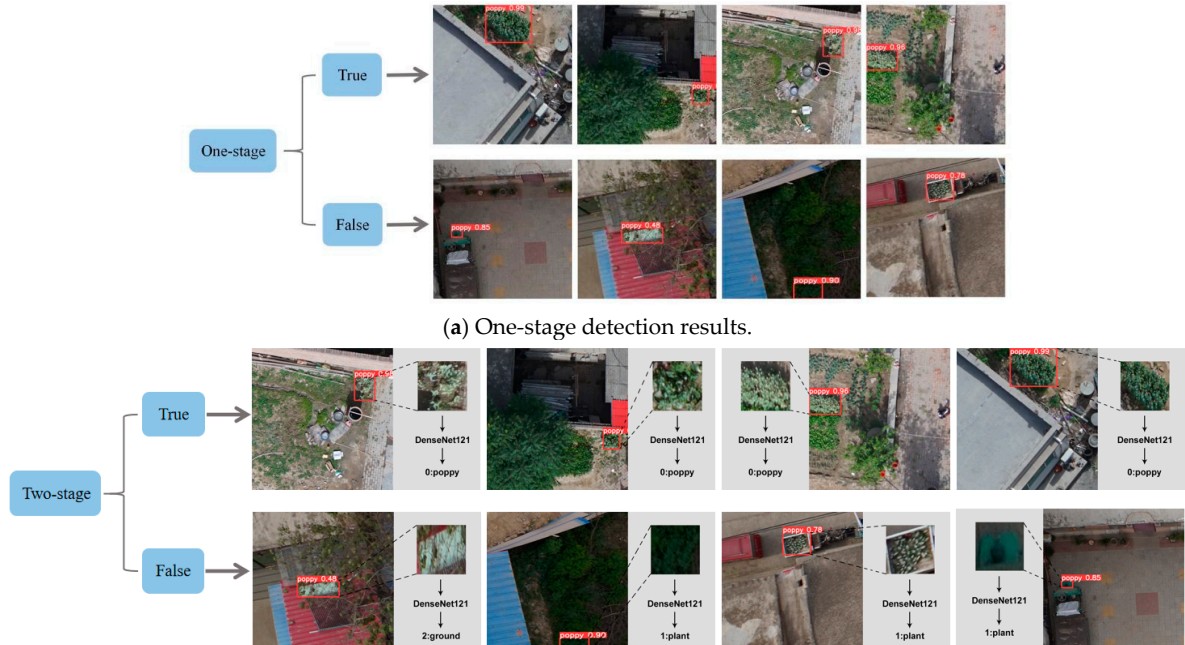

(**a**) One-stage detection results.

(**b**) Two-stage detection results.

**Figure 7.** One-stage and Two-stage detection results.

It can be seen from Figure 7 that the falsely judged *Papaver somniferum* images of the one-stage detector were subject to the problem of high confidence. The method of improving the confidence in the detection stage cannot solve this problem effectively. By applying the two-stage method, which means using a classifier to perform secondary judgment after the detector, the non-*Papaver somniferum* category with high confidence can be excluded so as to reduce the workload of subsequent manual screening. However, there were still some green crop images that are difficult to distinguish.

## 5. Discussion

(1) It is technically feasible to use target detection to monitor illicit opium *Papaver somniferum* cultivation. Researchers explored opium poppy detection in satellite images and low-altitude images of drones. The images detected in the literature [41] come from satellites, so the method in the paper is only effective for detecting large opium poppy cultivation fields. Zhou et al. [3] proved through experiments that reducing the flying height of a UAV can improve the detection accuracy because reducing the flying height can improve the ground resolution of pixels in the image and make the opium poppy features in the image clearer. However, low-altitude flight is easily affected by flight safety problems caused by tall buildings. Wang et al. [4] improved the original YOLOv3 by adding an ASPP module, replacing ordinary convolution with ResNext grouping convolution, and adding a wider range of detection frames to improve the accuracy of the model. However, the false positive rate of this method is high in practice. The two-stage method of detection and recognition proposed in this paper can effectively reduce the false positive rate. A comparison between our work and that of other authors is shown in Table 11.

**Table 11.** The comparison between our work and that of other authors.

| Methods | Stages | Height/m | Precision/% | Recall/% |
|---|---|---|---|---|
| Liu et al. [41] | one | 506,000 | 95.00 | 85.00 |
| Zhou et al. [3] | one | 30/60/150 | 96.37 | - |
| Wang et al. [4] | one | 120 | 94.30 | 90.70 |
| Our | two | 130–145 | 97.70 | 94.90 |

(2)  While collecting data and detecting *Papaver somniferum* from the images in the early stage, it was found that *Papaver somniferum* plants might be in different growth stages. In remote sensing images, the *Papaver somniferum* plants in the seedling stage, flowering stage and fruiting stage were of different characteristics. Specifically, the *Papaver somniferum* plants in the seedling stage were generally low in size, with the leaves in emerald color. The *Papaver somniferum* plants in the flowering stage had grown taller; the leaves became pale green with a serrated shape, and the flower color could be purple or white (the follower color of mutated *Papaver somniferum* might be more difficult to detect). In the fruiting stage, the leaves and fruits of *Papaver somniferum* were both in pale green color. The above factors make the "intra class difference" of poppy targets larger and improve the difficulty of detection. The key to solve this problem is to obtain as many *Papaver somniferum* samples as possible, so that the model can learn enough *Papaver somniferum* characteristics.

(3)  The plot area of opium poppy cultivation is closely related to the detection accuracy. However, due to the size of the data set, the functional relationship between detection accuracy and plot area is still unclear. In the future, we should collect more detection accuracy of different plots and establish a quantitative description function between plot area and detection accuracy.

## 6. Conclusions

The use of UAV has become the main approach to combat illicit *Papaver somniferum* cultivation. However, at present, the acquired images mainly rely on manual screening through naked eyes. In view of this, we proposed a two-stage method for the detection of *Papaver somniferum* cultivation based on target detection and classification assistance. The results show that the two-stage method proposed in this paper has the same recall rate of 95% with the one-stage method, but the false detection rate is reduced by 73.88%. This achievement greatly reduces the workload for subsequent manual screening.

**Author Contributions:** Q.W.: writing—original draft preparation; C.W. and H.W.: writing—review and editing; C.Z.: methodology; G.T.: supervision, data curation; Y.Y.: visualization, software; H.Z.: investigation, validation. All authors have read and agreed to the published version of the manuscript.

**Funding:** This research was funded by National Key Research and Development Program of China, grant number 2019YFD1101105, and in part by the National Natural Science Foundation of China, grant number 61871041, and in part by the Hebei Province Key Research and Development Program, grant number 20327402D, 19227210D.

**Acknowledgments:** We are grateful to our colleagues at Hebei Key Laboratory of Agricultural Big Data and National Engineering Research Center for Information Technology in Agriculture for their help and input, without which this study would not have been possible.

**Conflicts of Interest:** The authors declare no conflict of interest.

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
