# Peer review of "A Two-Stage Low-Altitude Remote Sensing Papaver Somniferum Image Detection System Based on YOLOv5s+DenseNet121"

_remotesensing, doi:10.3390/rs14081834_

Round 1

Reviewer 1 Report

A two-stage low-altitude remote sensing poppy image detection system based on YOLOv5s+DenseNet121

In this work the authors introduce  a two-stage method for detection of poppy cultivation sites, based on target detection and image classification. 

The work is in general interesting and well written, howeever improvements are needed, as listed below. 

2.1. Data acquisition

Please add pixel resolution and averag eground resolution for the collectede images.

It is not clear how ground truth data have been achieved: do the authors know the location of all poppy cultivations? I can guess the are cultivations that are out of the control of the authorities and not recognized by the UAV flights: such cases constitute false negatives FN that might decrease the claimed accuracy (Precision and F1).

2.2. Data processing

Author report that "appropriate image enhancement strategies need to be applied to improve the quality, color and contrast of the original images. In this paper, three enhancement methods were adopted, which are contrast enhancement, brightness enhancement and random angle rotation with brightness enhancement.". I cannot understand how a random angle rotation can be classified as an image enhancement strategy. 

As I understand, authors are doing data augmentation: if so, the actual variations (as can be seen by Table 1, columns a, b and c) are not so critical. Thereefore I wonder if they are really useful or are really increasing the accuracy of the approach. 

3. Experiment

As I understand, the paper make referencee to Papaver somniferum. Or also to Papaver bracteatum? Or also to other Papaveraceae? Please specify and add the scientific name into the title. 

Most of the error done by the detection and/or classification system might arise in the case of Papaveraceae species not of interest for the study. Different species (hard to distinguish from e.g. Papaver somniferum) might increase FP (and thus Recall and F1): have the authors considered different species in the paper (in the "plant" category? how many images?) 

4. Experiment and Analysis

I think that the performance is afefcted by the average size of thee plantation: I can guess that bigger planted areas can be more easily and effectively detected by the proposed approach. Also for the method it is not so critical not recognizing a single plant, but conversly would be critical not recognizing a big cultivated area. I think the authors should discuss the performance as a function of the size of the planted area. 

Also not all of the poppy plants grow at the same time: insome cases senescence is starting earlir or later, based on sun radiation, or shadowingm, or water stress, .... Authors has mentioned this in the paper, but should discuss the issue in a deeper form. 

I guess that better detction is allowed by highr ground resolution, which on th other hand can cause an incrase in processing time. Please give moree details about training and processing time. 

Other:

Please report the meaning of acronyms CBL FPN and PAN. 

In case the paper will be accepted for revision, please address above comments and correct accordingly the paper,

- giving your pertinent comments in the “Response to reviewer” document

- reporting in the “Response to reviewer” document also the paragraph with amended text highlighted with yellow colour or the new amended figure.

Reviewer 2 Report

This is a current and relatively interesting manuscript, which falls to the scope of Remote Sensing scientific journal. Manuscript deals with the detection of illegal poppy fields. The authors use the YOLO machine learning object detection architecture, which is part of the Python libraries.

However, while reading the manuscript, I came across some shortcomings that should be eliminated.

Introduction

Other uses of poppy seeds in the kitchen should be mentioned here. In our country it is commonly used in baking. I would also welcome a reference to the information given on p. 1, l. 39-40.

The main aim of this article should be clearly defined at the end of the Introduction section.

Materials

Please avoid repeating information, e.g. p 4, l. 163 and p. 2, l. 178 (about 643 images). I do not fully understand the statement on p. 4, l. 167 that "The collection of poppy images is very challenging ....". So, what is the advantage of this poppy field search system over the practices used so far?

Experiment (Results?)

Figures 3 and 4 are so small that they are illegible.

Experiments proved weak learning ability for small targets (p. 11, l. 362-363). Does this mean that small poppy fields will be difficult to detect? What is the proportion of these small sized fields? What is the size of a sufficiently large sample (p. 13, l. 395-396)? Please be specific, I think this is fairly crucial information.

Discussion

Discussion with the results of other authors is missing. There is also no reference to literature. I consider this to be a major shortcoming. It is not clear whether the authors were better or worse in their experiments compared to the results already published.

Conclusion

Conclusion is too long. Please only indicate what were the main results of your research that were not yet known. As I understand it, 19 of the 20 poppy fields were correctly detected, and in addition another 291 or 76, which were additionally detected and were not poppy fields. I do not consider this to be a completely encouraging result. In conclusion, I also lack essential information about how large the poppy field must be in order to be reliably detected.

Reviewer 3 Report

This paper uses a two-stage approach (target detection and image classification) on poppy plants, compared with the one-stage method, The correctly detected parts have the same good performance, the difference is that the wrongly detected parts can be Reduced from 291 to 76, a reduction of nearly 73.88%.

(1) The authors should
compare YOLOv5s with other different models, and more models (ex: Mask-RCNN, DSSD) should be added for comparison to prove that it turns out that YOLOv5s is the better method.
(2)
When comparing DenseNet121 with other different models, more models should be added (ex : EfficientNet, Xception, InceptionResNet) for comparison to prove DenseNet121 is better.
(3)
How does using a two-stage approach reduce the number of false detections?

Round 2

Reviewer 1 Report

Authors have spent big efforts to respond to the referee, but have done only minor corrections to the paper.

REFEREE: 

It is not clear how ground truth data have been achieved: do the authors know the location of all poppy cultivations? I can guess the are cultivations that are out of the control of the authorities and not recognized by the UAV flights: such cases constitute false negatives FN that might decrease the claimed accuracy (Precision and F1).

AUTHORS: 

The process of acquiring and labeling data is as follows: firstly, we collect the original images through the UAV and record the GPS position of each image. Secondly, the professionals manually screen these images and select all the
images of suspected poppy, and finally the supervisors go to the scene according to the GPS information to verify whether the images of suspected poppy are real poppy. If it is a real poppy target, it is marked as a ground truth sample. Therefore, the training set will neither omit the real poppy as a false negative sample, nor contain a false positive sample that is not a poppy target.

REFEREE: 

Explanation has not ben added to the paper.

Additionally the authors have now explained that the ground resolution is 2 cm: therefore a flower is characterized on avereage by 4-8 pixels. Even though the supervisors have a lot of experience, I think it is almost impossiblee to recognize all of the poppy flowers within several squared chilometres. So FLASE NEGATIVES are certainly present or anyhow the contrary is not demonstrated in the paper. 

REFEREE: 

[...] I cannot understand how a random angle rotation can be classified as an image enhancement strategy. 

AUTHORS: 

When UAV collects images, it is necessary to set a certain side overlap between routes to ensure that the image acquisition can completely cover the whole survey area. Although the overlapping part contains the ground at the same position, the ground at the same position is not exactly the same in the two images. The ground at the same position often has a certain deflection angle in the two images. In addition, in the process of image acquisition, it is inevitable to be affected by the change of illumination. The poppy images are different in contrast and brightness. Based on the above reasons, we take random angle rotation and brightness enhancement as the image data augmentation. Experiment results show that this enhancement strategy is really useful and really increases the accuracy of the approach.

REFEREE: 

Please add explanation in the paper. 

REFEREE: 

As I understand, the paper make referencee to Papaver somniferum. Or also to Papaver bracteatum? Or also to other Papaveraceae? Please specify and add the scientific name into the title. 

AUTHORS:

Thanks for your comments. Based on the regulatory investigation, the unauthorized cultivation of opium poppy is mainly used as the raw material  of psychotropic drugs, and the main species is Papaver somniferum L. No other poppy species have been found.

REFEREE: 

The title has not been corrected. 

REFEREE: 

I think that the performance is affected by the average size of the plantation: I can guess that bigger planted areas can be more easily and effectively detected by the proposed approach. Also for the method it is not so critical not recognizing a single plant, but conversly would be critical not recognizing a big cultivated area. I think the authors should discuss the performance as a function of the size of the planted area. 

AUTHORS: 

The size of planted areas is one of the factors affecting detection performance. Experiments show that the detection accuracy of larger fields is higher, and the detection of smaller fields is more difficult. As far as we know, there is no research on the relationship between field area and model detection performance.
Therefore, establishing the functional relationship between field area and model detection performance is our focus in the future. The reality is that the target characteristics of largescale cultivation of opium poppy are obvious and easy to be detected by regulatory authorities. On the contrary, scattered and small-area planting is not easy to be found by the regulatory authorities because of its insignificant characteristics, which has become the focus and difficulty of the regulatory work. We have updated the discussion section.

REFEREE: 

The discussion has been changed but not including the issue mentioned here. If experiments show that the detection accuracy of larger fields is higher, than please add a graph (e.g. showing accuracy vs size) and a discussion on this.

REFEREE: 

Also not all of the poppy plants grow at the same time: in some cases senescence is starting earlir or later, based on sun radiation, or shadowingm, or water stress, .... Authors has mentioned this in the paper, but should discuss the issue in a deeper form. 

AUTHORS: 

No answer was given and no correction was accordingly provided in the paper. 

REFEREE: 

Please consider the comment. 

REFEREE: 

I guess that better detection is allowed by higher ground resolution, which on the other hand can cause an incrase in processing time. Please give more details about training and processing time. 

AUTHORS:

The training duration of YOLOv5s used in this paper is 130 minutes, the training duration of densennet121 is 21 minutes, and the total training duration is 151 minutes. The average time of processing a picture by the two-stage method is less than 0.02s.

REFEREE:

The comment was related to the time as a function of resolution. 

In case the paper will be accepted for revision, please address above comments and correct accordingly the paper,

- giving your pertinent comments in the “Response to reviewer” document

- reporting in the “Response to reviewer” document also the paragraph with amended text highlighted with yellow colour or the new amended figure.

not corrcted acc

Reviewer 2 Report

The authors responded satisfactorily to all my comments and incorporated it into the manuscript. Manuscript was improved.

Reviewer 3 Report

Discussion and conclusion parts should be expanded. Please explain the results in tables 7-8 with the new experiments. Provide the advantages and disadvantages of the proposed method and other state of art methods in tables.  Also, English should be checked again by the native speaker. Reference can be add more about deep learning and some other related studies. Please make more efforts on the revised version. 
